# Identification of Putative Candidate Genes from *Galphimia* spp. Encoding Enzymes of the Galphimines Triterpenoids Synthesis Pathway with Anxiolytic and Sedative Effects

**DOI:** 10.3390/plants11141879

**Published:** 2022-07-20

**Authors:** Dianella Iglesias, Marcos de Donato Capote, Alfonso Méndez Tenorio, Ana Victoria Valdivia, Claudia Gutiérrez-García, Sujay Paul, Hafiz M. N. Iqbal, María Luisa Villarreal, Ashutosh Sharma

**Affiliations:** 1Centro de Investigación en Biotecnología, Universidad Autónoma del Estado de Morelos, Cuernavaca 62210, CP, Mexico; dianella.iglesiasrod@uaem.edu.mx; 2Centre of Bioengineering, NatProLab, School of Engineering and Sciences, Tecnologico de Monterrey, Querétaro 76130, CP, Mexico; mdedonate@tec.mx (M.d.D.C.); victoria.valdivia@tec.mx (A.V.V.); a00887520@itesm.mx (C.G.-G.); spaul@tec.mx (S.P.); 3Escuela Nacional de Ciencias Biológicas, Instituto Politécnico Nacional, Ciudad de Mexico 56565, CP, Mexico; amendezt@ipn.mx; 4School of Engineering and Sciences, Tecnologico de Monterrey, Monterrey 64849, CP, Mexico; hafiz.iqbal@tec.mx

**Keywords:** functional annotation, *Galphimia* spp., transcriptome

## Abstract

*Galphimia* spp. is popularly used in Mexican traditional medicine. Some populations of *Galphimia* exert anxiolytic and sedative effects due to the presence of the modified triterpenoids galphimines. However, the galphimine synthesis pathway has not yet been elucidated. Hence, in this study, a comparative transcriptome analysis between two contrasting populations of *Galphimia* spp., a galphimine-producer, and a non-galphimine-producer, is performed using RNA-Seq in the Illumina Next Seq 550 platform to identify putative candidates genes that encode enzymes of this metabolic pathway. Transcriptome functional annotation was performed using the Blast2GO in levels of gene ontology. For differential expression analysis, edgeR, pheatmap, and Genie3 library were used. To validate transcriptome data, qPCR was conducted. In producer and non-producer plants of both populations of *Galphimia* spp., most of the transcripts were grouped in the Molecular Function level of gene ontology. A total of 680 differentially expressed transcripts between producer and non-producer plants were detected. In galphimine-producer plants, a larger number of highly expressed transcripts related to acyclic and polycyclic terpene synthesis were identified. As putative candidate genes involved in the galphimine synthesis pathway, P450 family members and enzymes with kinase activity were identified.

## 1. Introduction

*Galphimia* spp. is a medicinal plant widely used in Mexico to treat various ailments, including fever, labor pains [1], diarrhea [2], and anxiety, due to its anti-inflammatory and anxiolytic properties [3,4]. The anxiolytic and sedative effects of the extract have made this plant the target of pharmacological [5,6,7], phytochemicals [3,8,9,10], biotechnological [6,11,12], metabolomic [13,14] and genetics studies [15,16]. Nevertheless, the effects of some populations of the plant, are due to the presence of the modified triterpenoid galphimines [17].

Different metabolomic studies have already been carried out to detect the presence of galphimines as well as their concentration among six Mexican populations of *Galphimia* spp. [13,14], from which only two (located in Jalpan, (Querétaro) and Dr. Mora (Guanajuato)) were identified as galphimines producers. Moreover, to identify the individual populations of *Galphimia* a DNA barcode study was conducted by Sharma et al. [15], and the clades obtained showed clear differences between the galphimine-producer populations (Doctor Mora and Jalpan) and the other analyzed populations. In another study, conducted by Gesto-Borroto et al. [16], among nine populations of *Galphimia* spp. evaluated by DNA barcode, four galphimine-producing populations were identified, including Dr. Mora (Guanajuato) and Jalpan (Querétaro), corroborating the previous studies. 

In several plants, stages of triterpene synthesis have been studied [18,19,20]; however, the galphimine-producing pathway in *Galphimia* spp., is still elusive. Nevertheless, a biogenetic proposal for its production has been published from the oxidation and decarboxylation of the Taraxasteryl cation [21], although it has not yet been demonstrated. 

Regarding the biological properties of the galphimines, the hypothesis to be tested was the following: „If a transcriptomic study and a comparison are carried out in two populations of *Galphimia* spp.; a gaphimine producer population (GP) with a non-galphimine producer population (NPG), the putative candidates genes, that encode the enzymes involved in the production pathway of these triterpenoids, can be identified. Hence, this paper aimed to analyze transcriptomes from a GP and NPG populations of *Galphimia* spp., to identify putative candidates genes that encode enzymes for the synthesis of these triterpenoids.

## 2. Results

### 2.1. Plant Material

Plants with phenotypic characteristics similar to those described for the *Galphimia* genus [22], were obtained from both populations GP and NPG. They were characterized by having several adventitious roots, 5–6 cm long, with abundant adsorbent hairs. They are 1.4 m tall shrubs, with approximately 15 lateral branches in the GP plants and 10 in the NGP plants. The leaves were elliptical, with prominent secondary veins and their area was 13.75 cm^2^. Yellow inflorescences were observed arranged in terminal racemes.

### 2.2. Transcriptome Sequencing

RNA samples had a high quality with RNA integrity number (RIN) ≥8. Around 56.6 Gb of clean data were obtained from transcriptome sequencing. Q30 percentages of clean data for all samples were 97.2%, and the Guanine/Cytosine (GC) were rated between 50 to 51%. The total bases per sample obtained were in a range between 2,704,225 and 675,148.

After FastQC analysis, it was observed that the quality of bases per sequence was optimal since all readings were located in the green area of the graphic. The fast QC analysis showed that the filtered reads had quality valius >Q30, but the reads by sample GC content and duplication levels differed. After trimming the reads with Trimmomatic, these metrics reached the standard values. Post-assembly, around 9000 transcripts from leaf samples and 5000 transcripts from roots samples were obtained (Table 1).

### 2.3. Transcriptome Functional Annotation

The gene ontology analysis showed a different number of annotated genes: 5900 in GP plants, 3600 in the NGP ones, and 1500 in the controls. In GP plants and controls, most of the transcripts were grouped in gene ontology categories MF and CC. In the NGP plants, most of the transcripts were grouped mostly in the MF and BP categories. At the level of gene ontology BP, the largest number of transcripts were associated with cellular processes, biological processes, and biological regulation (Figure 1), while at the MF level, most of the transcripts annotated were related to catalysis, binding, and transporter. In the CC, transcripts of three samples are located as a cellular anatomical entity and protein-containing complex.

According to the enzyme distribution code in the GP population, several transcripts were identified (Figure 2), such as 1300 with transferase activity, 1020 with hydrolase activity, and 810 with oxidoreductase activity; while in the NGP population transcripts were observed as follows: 2250 with oxidoreductase activity, 2220 with transferase activity, and 2050 with translocase activity. In control, 270 transcripts with hydrolase activity, 260 with transferase activity, and 250 with translocase activity were observed.

### 2.4. Differential Expression Analysis

When comparing the GP population with the NGP population, a total of 680 transcripts with differential expressions were found. Of these, 313 were positively regulated and 367 negatively regulated. The volcano plot shows that both positively regulated (represented by red dots) and negatively regulated transcripts (represented by green dots) have a large-magnitude logFC due to the significant separation from the center, and a high level of statistical significance since the location of dots over 100 of log10 (PValue). The greatest statistical significance was observed in positive regulated transcripts. (Figure 3a).

A total of 203 transcripts were differentially regulated between GP plants and control. The number of negatively regulated transcripts (logFC < 2) was 116, higher than positively regulated transcripts (logFC > 2), which was 87 in number. The positively regulated transcripts presented a higher statistical significance (Figure 3b). In addition, there were changes in the logFC and high values of -log10 (PValue), indicators of the statistical significance of some transcripts.

In the GP plants, a greater number of differentially expressed transcripts with positive regulation were observed as compared to the NGP and the control (Figure 4), suggesting a high metabolic activity in these plants. Moreover, the transcripts with the highest expression values in the GP plants were mostly hypothetical proteins, ribosomal proteins, transferases, endonucleases, and carboxypeptidases. However, in NGP plants, the transcripts with the highest expression encoded mitochondrial receptors, ribosomal proteins, enzymes with synthase activity, and polygalacturonases.

In the comparison between GP plants and controls, the transcripts with the highest expression values in the GP plants were ribosomal proteins, proteins associated with senescence, and enzymes with endonuclease, polymerase, and dehydrogenase activities (Figure 5). In controls, the transcripts with higher expression values code for receptors, aquaporins, chaperones, endonucleases, and methyltransferases.

### 2.5. Identification of Putative Candidate Genes That Encode Enzymes of the Galphimine Synthesis Pathway

Based on the results of functional annotation, 44 transcripts associated with the terpenes synthesis pathway were identified in GP plants (Table 2). Out of those, two transcripts encode for 3-hydroxy-3-methylglutaryl-CoA reductase (a regulatory enzyme of the mevalonic acid pathway), and 13 transcripts encode enzymes of the methylerythritol phosphate pathway. Among other transcripts, three encode for enzymes associated with intermediates in the isoprenoid biosynthesis, one related to terpenes synthesis, and 25 with triterpenes.

In the NGP plants, 35 transcripts were found associated with terpene synthesis, among which one encodes the regulatory enzyme of the mevalonic acid pathway, 10 associated with methylerythritol phosphate pathway, five with the intermediary in the isoprenoid biosynthesis, three transcripts related to terpenes synthesis and 16 with triterpenes (Table 2). In the control, the following transcripts were observed: one that encodes for 3-hydroxy-3-methylglutaryl-CoA reductase, one for P450, and five for terpene synthase.

The GP plants presented a greater number and diversity of transcripts related to the synthesis of terpenes with high values of gen counts compared to the NGP plants. This suggests a higher metabolic activity in the terpene synthesis pathway in these plants. P450 TBP, P450, and terpene synthase were the enzymes with the highest gen count. In the NGP plants, the enzymes with the highest gen counts were terpene synthase and CYP82G1.

In the co-expression network, 22 nodes were observed (Figure 6), among which, HMGCR (3-hydroxy-3-methylglutaryl-CoA reductase), IPPI (isopentenyl diphosphate isomerase), and FT (farnesyl transferase) showed the highest number of interactions with other nodes. The expression of the majority of P450 family members was closely related.

### 2.6. Validation of Transcriptome Data by qPCR

By qPCR analysis, the results of the transcriptomic study were validated. The expression levels of putative candidate genes related to galphimine synthesis were found to be similar in both differential gene expression analysis (Table 2) and qPCR (Figure 7). The expression was higher in GP plants because CT values were lower than in NGP plants (Figure 8) and significant differences between plants of both populations in fold change value were observed (Figure 6). In GP plants the genes with a higher level of relative expression were P450 TBP, 5’-AMP-activated protein kinase, and P450. Overall, the relative expression of these genes in NGP plants was low.

## 3. Discussion

*Galphimia* spp, is an important medicinal plant in Mexico, with several applications in traditional medicine; thus, the main investigations with this plant are based on its pharmacological effects, especially for the modified triterpenoids galphimines. However, genes that encode enzymes of the galphimines synthesis pathway are unknown. Nevertheless, identifying new genes is a complex and great important process, especially when the genome is not available. The least complex and economical strategy is the use of omics tools such as transcriptome and digital gene expression profile analysis. These strategies allowed to explore and analyze genes with a differential expression which increases the chance of discovering new genes as well as the characterization of their structure. In this study, the Illumina Next seq 550 platform has been used to investigate the transcriptomes of *Galphimia* spp, which allowed the successful identification of several galphimine synthesis-related genes.

The GO annotation provides a preliminary indication of the nature of a gene product. In this study, most of the transcripts have been recorded at the MF and CC gene ontology levels, which provided an approximation to the elementary functions in which the transcripts participate as well as the parts of the cells or extracellular environment they’re located in. Given that galphimines are triterpenoids of polycyclic structure and that the synthesis of these metabolites includes oxidation and transfer of functional groups, it is predicted that most of the transcripts identified are associated with transferase and oxidoreductase activities.

Triterpenes are one of the most important groups of secondary metabolites in plants. These are generally produced from the acyclic 30-carbon precursors 2,3-oxidosqualene in eukaryotes. The conversion of 2,3-oxidosqualene to various cyclic triterpenes by squalene cyclase is the first step in the diversification of the biosynthetic pathway of the triterpenes. Subsequently, the generated intermediaries undergo a large number of modifications regio and stereospecific structures catalyzed by cytochrome P450 (CYP), acetylases, and UDP glucosyltransferases that catalyze oxidation, methylation, acetylation, malonylation, and glycosylation, [23]. Plant CYPs transfer two electrons from cofactors NADPH and atmospheric oxygen to their substrates. Generally, plants require a CYP reductase to support electron transfer [24]. The oleanane, derived from β-amyrin, is one of the largest representatives of triterpene scaffolds pentacyclic. Approximately 55 CYPs act on scaffolds of pentacyclic triterpenes of plants. Most of them belong to members of the CYP716 family, although other families such as CYP51, CYP71, CYP72, CYP87, CYP88, and CYP93 also participate in modifications of pentacyclic triterpenes [23].

Members of the cytochrome P450 family are involved in the synthesis of triterpenes. Geisler et al. performed a functional analysis of CYP51 in *Nicotiana benthamiana* leaves and demonstrated that this enzyme can catalyze the hydroxylation and the epoxidation of simple triterpene β-amyrin to give 12,13β-epoxy-3β, 16β-dihydroxy-oleanane (12, 13β-epoxy-16β-hydroxy-β-amyrin) [25].

Also, three oxidosqualene cyclases (OSC) TwOSC1, TwOSC2, and TwOSC3 were isolated and characterized from *Tripterygium wilfordii*. TwOSC1 and TwOSC3 were multiproduct friedelin synthases and were involved in celastrol biosynthesis, while TwOSC2 was a β-amirin synthase [26]. In that work, the authors proposed a biogenetic pathway for celastrol in which hydroxyfriedelanes synthesis was mediated by P450 [26]. On the other hand, in *Aralia elata*, CYP716A295 and CYP716A296 were identified as the candidate genes most likely associated with oleanolic acid synthesis [27].

Galphimines are modified triterpenoids, classified as Nor-secofriedelanes. These are derived from oleanane and later from friedelin. The arrangements in the intermediates between friedelin and norsecofriedelanes involve oxidation-reduction reactions for opening a ring and the C3-C4 bond break, resulting in the two functional groups required for the formation of the seven-membered lactone characteristic of galphimines. Considering the functions that have been reported for the P450 family in the synthesis of triterpenes and the gene count of the transcripts encoding members of the P450 family in the GP population (Table 2), as well as the absence of many of these in the NGP population, these enzymes could be considered potential candidates for galphimines synthesis, as suggested in Figure 9.

In differential expression analysis, the GP plants presented a greater number of genes related to the synthesis of terpenes as compared to the NGP plants. In both analyzed populations, some transcripts encode the mevalonic acid and methylerythritol phosphate pathways as well as precursors of the biosynthesis of isoprenoids, terpenes, and triterpenes. However, in the GP plants, the number and diversity of identified enzymes were higher. Since terpenes are a group of secondary metabolites that are highly represented in plants [29], they are present in both studied populations. However, the GP plants presented a higher expression of the enzymes of this pathway and, as a consequence, a higher content of these metabolites. This corroborates with the results of the study by Sharma et al. where the chemical profile of the *Galphimia* spp., the populations used in this study was analyzed by thin-layer chromatography (TLC), suggesting the presence of galphimines in the GP population, while other terpenes were in the NGP population [15].

Among the enzymes with the highest levels of expression in both populations were terpene synthase and CYP82G1. This is expected because terpene synthase diversifies isoprenoid precursors into a large number of mono and sesquiterpenes [30]. CYP82G1 participates in the synthesis of volatile homoterpenes. These homoterpenes are considered among the most common plant compounds that act as part of defense mechanisms against herbivores [31].

Co-expression networks are a new resource for understanding convergent pathways and their relations. This resource has been used to address various biological questions [32]. In the present study, the enzymes selected as possible candidates for the galphimines synthesis pathway interacted closely with each other, as well as with other enzymes involved in stages before the triterpenes synthesis. It has shown the genes that participate in the same metabolic pathway or those are under the same regulatory mechanism. It is worth remarking on the interaction of 3-hydroxy-3-methylglutaryl-CoA reductase, with a considerable number of the identified enzymes that are associated with the synthesis of polycyclic triterpenes. This enzyme is a regulator of the mevalonic acid pathway, which seems to be the proper route for synthesizing triterpenes [33]. The 3-hydroxy-3-methylglutaryl-CoA reductase seems to control the core of the terpene’s metabolism in the cytosol, regulating the production of the compound obtained, as well as the activity of the gene products involved in these reactions.

In this study, relative expression determined by qPCR corroborated the results from the transcriptomic analysis. Our results allow us to support that the genes selected as candidates involved in galphimine biosynthesis are present in high concentrations in the GP plants and are in low concentrations or absent in the NGP plants.

Given the limited availability of reference genomes in plants, the transcriptomic analysis of bioactive metabolites in producer and nonproducer plants is a valuable strategy to detect the genes involved in the synthesis pathways of the compounds of interest.

## 4. Material and Methods

### 4.1. Plant Material

The collection of plants and seeds was carried out manually under the permission of Secretaría de Medio Ambiente y Recursos Naturales (SEMARNAT) in a GP population, Doctor Mora (Guanajuato) (W 100° 19.22; N 21° 08.7; 2120 m altitude) and in a NGP population, Tepoztlán (Morelos) (W 99° 06.97 N 18° 59.35, 1700 m altitude). The samples were deposited in the HUMO Herbarium of the Sierra de Huatla Environmental Education and Research Center of the Universidad Autónoma del Estado de Morelos (UAEM) to be properly identified, voucher numbers 15189 and 15485, respectively. From 10 seeds collected from each population, *Galphimia* spp. seedlings were obtained and cultivated at 20 °C of temperature and 60% of humidity for 2 months and then in natural conditions for 15 months. The experiment was performed from September 2019 to January 2021 in the Tissue Culture Lab at the Bioengineering Center of Instituto Tecnológico y de Estudios Superiores de Monterrey, Querétaro, México. During the cultivation period of the plants, parameters related to their growth and development were evaluated, such as height, number of lateral branches, leaf area by ellipse equation, and appearance of inflorescences.

### 4.2. RNA Extraction, Library Preparation, and Transcriptome Sequencing

Total RNA was extracted using RNeasy Plant Mini Kit (QIAGEN, Hilden, Germany) from the leaves of plants originating from both populations as well as from the roots of the plants of the producer population as a control. The quality and integrity of the total RNA were evaluated using a Qsep 100 Advance, Bioptic. RNA concentration was measured using a NanoDrop 2000 spectrophotometer (NanoDrop Technologies, Technologies, Wilmington, DE, USA) and a Qubit 4^TD^ fluorometer (Invitrogen, Thermo Fisher, Waltham, MA, USA).

Six RNA-Seq libraries (leaves samples from a GP population plant, leaves samples from the NGP population, root samples from the GP population, and one replica of each sample) were constructed using the Illumina TruSeq Stranded Total RNA protocol. Roots were used as a control since the production of galphimines is negligible in this organ. The quantity and quality of enriched dscDNA were assessed using a Qubit fluorometer (Thermo Fisher, Waltham, MA, USA) and Agilent 2100 Bioanalyzer (Agilent Technologies, Santa Clara, CA, USA). The libraries were sequenced using the Illumina Next Seq 550 platform; for 2 × 76 sequencing, cycles to obtain paired-end readings.

#### Reads Quality Control and De Novo Transcriptome Assembly

The quality of the reads obtained by sequencing was analyzed using the FastQC software (v.0.11.8) and Trimmomatic (v.0.38) tool was used to trim the sequence reads. Afterward, de novo transcriptomes assembly was carried out, for each sample, by Trinity (v.2.9.1) using the default settings mode.

### 4.3. Transcriptome Functional Annotation

Transcriptome functional annotation was performed using the Blast2GO (v.4.1.9). The assembled transcripts were aligned by a tblastx with the NCBI database. A BLAST expectation value of 1.0 × 10^−3^ was applied, and for each sequence, 20 alignments were made. A sequence mapping was conducted, followed by functional annotation in levels of gene ontology Biological Process (BP), Molecular Function (MF), and Cellular Component (CC). Sequences that were not annotated by gene ontology were identified by InterPro, UniProt, and TAIR databases.

### 4.4. Differential Expression Analysis

To identify corresponding gene pairs between individuals in the two populations of *Galphimia* spp., Corset programs (v.1.09) (clustering de novo assembled transcripts) was used. After transcripts were compared and grouped, results were also verified by functional annotation. An abundance quantification of the transcripts assembled in each transcriptome was conducted using the Kallisto program (v.0.46.0.4). Then, the counting matrix was built in R (v.4.0.4) for Windows, with the use of the plyr, dplyr, and stats libraries. Genes with low library counts were filtered using the CPM Filter on a per-million count (CPM) basis. For the analysis of differential expression, the edgeR library was used. Samples in the counting matrix were normalized on a transcript-per-million (TPM) basis. Transcripts expression in the leaves of the GP plants was compared with transcripts of the NGP and with the control. The differentially expressed transcripts were considered to be those with a false positive rate (FDR) <0.01 and classified as positively regulated when log fold Change (logFC) > 2 and negative when logFC < −2. The transcripts with the highest differential expression values in each sample were visualized on a heat map constructed with the pheatmap library.

### 4.5. Identification of Putative Genes Related to the Galphimines Synthetic Pathway

Data obtained from the functional annotation of transcriptomes and the results of the differential expression analysis were used to identify genes involved in the synthesis of terpenes and putative candidate genes for the synthesis of galphimines. To identify the interactions between the genes related to the synthesis of terpenes and the possible candidate genes for the synthesis of galphimines, a co-expression network was also built, using the libraries of R, GENIE3, igraph, RCy3, and Rgraphviz. The co-expression network was calculated in the form of a weighted adjacency matrix, using ensembles of regression trees. Candidate regulatory genes were not used. The tree method was Random Forests. The number of candidate regulators randomly selected at each tree node was the square root of the total number of candidate regulators. The number of trees in each target was 1000. The co-expression network was visualized by Cytoscape (v.3.8.2).

### 4.6. Validation of Transcriptome Data by qPCR

cDNA of GP plants and NGP plants was obtained using the PrimerScript RT Reagent Kit with gDNA Eraser (TaKaRa, San Jose, CA, USA). The concentration and purity of cDNA were evaluated by Nano-Drop 2000 spectrophotometer (NanoDrop Technologies, Technologies, Wilmington, DE). According to transcriptome data of GP plants, nucleotide sequences of ten putative candidate genes related to galphimines synthetic pathway (CYP82C4, P450 TBP, PLAC8, 5’-AMP-activated protein kinase, serine/threonine-protein kinase AtPK2/AtPK19, P450, CYP71A24, CYP71B34, CYP72A, CYP81D), were used for primers designing (Table 3) by Primer3 plus program. The quantitative PCR was conducted using the TB Green^®^ Advantage^®^ qPCR Premix (Takara Bio USA, Inc., San Jose, CA, USA) following the manufacturer’s protocol in a StepOne™ Real-Time PCR System (Applied Biosystems, Waltham, MA, USA). The amplification conditions used were as follows: activation/denaturation at 95 °C for 10 s, 45 cycles of denaturation at 95 °C for 5 s, and annealing/extension at 60 °C for 30 sec followed by a melting curve analysis ranging from 56–95 °C. Three biological replicates with three technical replicates were used for each sample. The relative expression of the RNA was quantified by the 2^−ΔΔCt^ method using U6 as the reference gene [34].

## 5. Conclusions

Transcriptome analysis of two populations of *Galphimia* spp. allowed the detection of differences between a galphimine producer GP plant and a non-galphimine producer NGP plant. The higher expression of P450 family members and kinases in GP plants and their well-known role in triterpene synthesis make them putative candidates to encode enzymes involved in the galphimine synthesis pathway.

## Figures and Tables

**Figure 1 plants-11-01879-f001:**
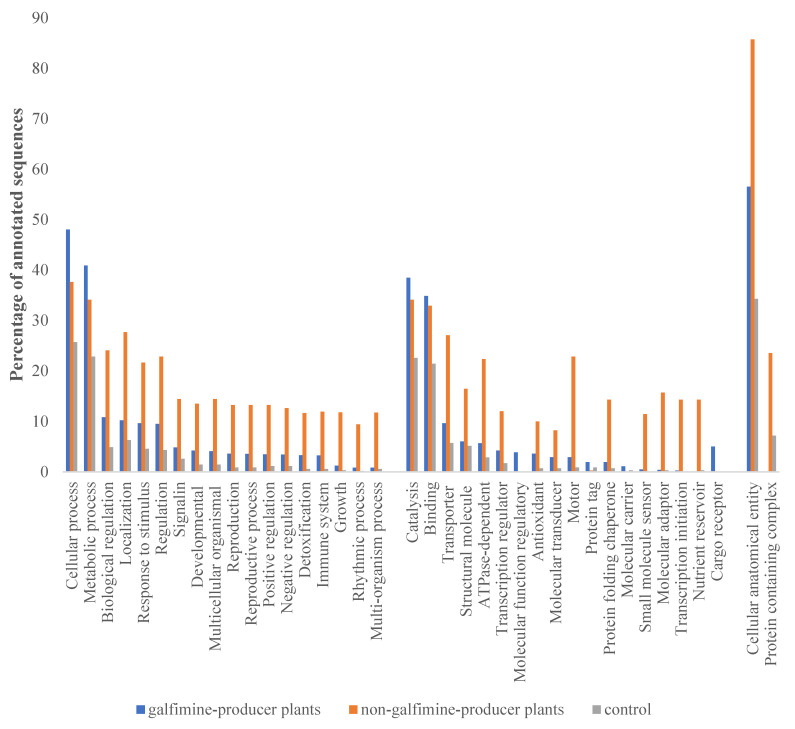
Percentage of annotated sequences of galphimine-producer plants (leaves samples, non-galphimine producer plants, (leaves samples) and control (roots samples) in gene ontology levels Biological Process (BP), Molecular Function (MF), and Cellular Component (CC).

**Figure 2 plants-11-01879-f002:**
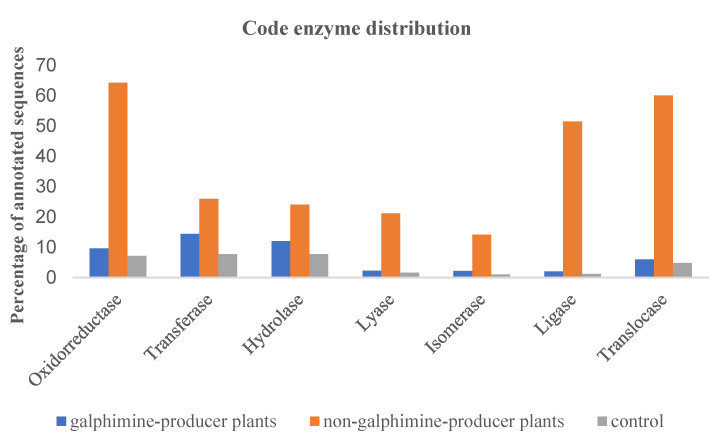
Enzyme distribution code of annotated sequences (percentage) in galphimine producer plants (leaves samples), non-galphimine producer plants (leaves samples), and control (roots samples).

**Figure 3 plants-11-01879-f003:**
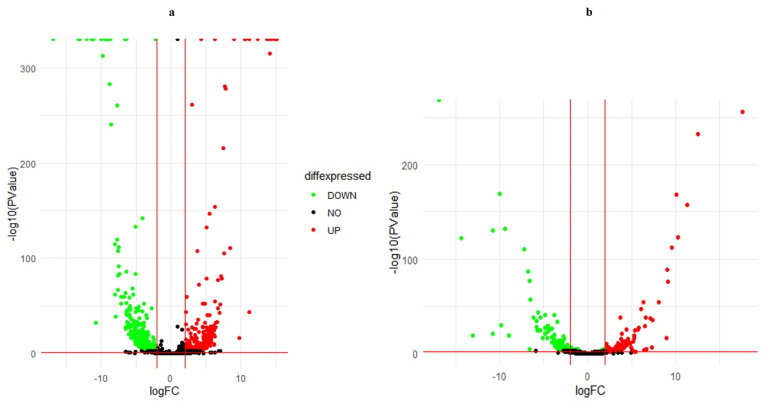
Volcano plot of differentially expressed genes (**a**) galphimine producer plants (leaves samples) versus non-galphimine producer plants (leaves samples) (**b**) galphimine producer plants (leaves samples) versus control (roots samples). X-axis and y-axis represent log2 fold-change differences between the compared samples and statistical significance as the negative log of differentially expressed genes *p*-values, respectively. The significantly up-regulated and down-regulated genes are indicated with red and green dots, respectively, while non-significant genes are shown as black dots.

**Figure 4 plants-11-01879-f004:**
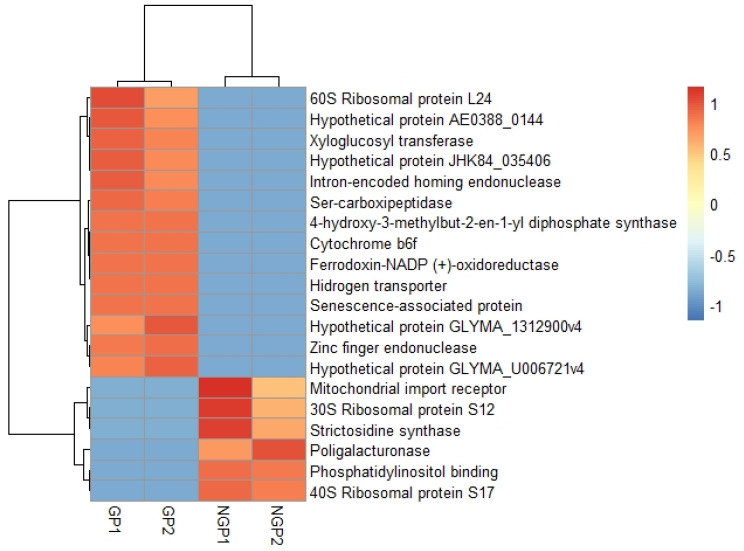
Comparison of transcripts with higher differential expression values between two populations of *Galphimia* spp. GP1, GP2 (leaves of galphimine producer plants). NGP1, NPG2 (leaves of non-galphimine producer plants). High values are represented in red color, the lowest values are represented in blue.

**Figure 5 plants-11-01879-f005:**
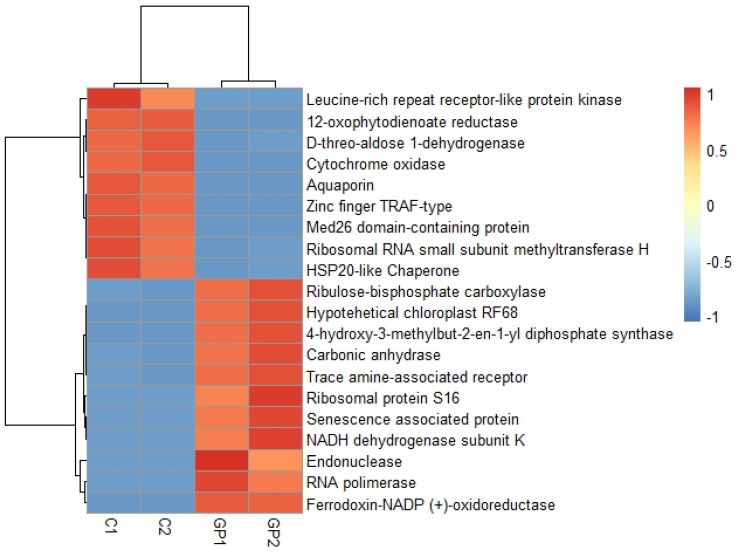
Comparison of transcripts with higher differential expression values between leaves and roots of galphimine producer plants. C1, C2 (roots). GP1, GP2 (leaves). High values are represented in red color, the lowest values are represented in blue.

**Figure 6 plants-11-01879-f006:**
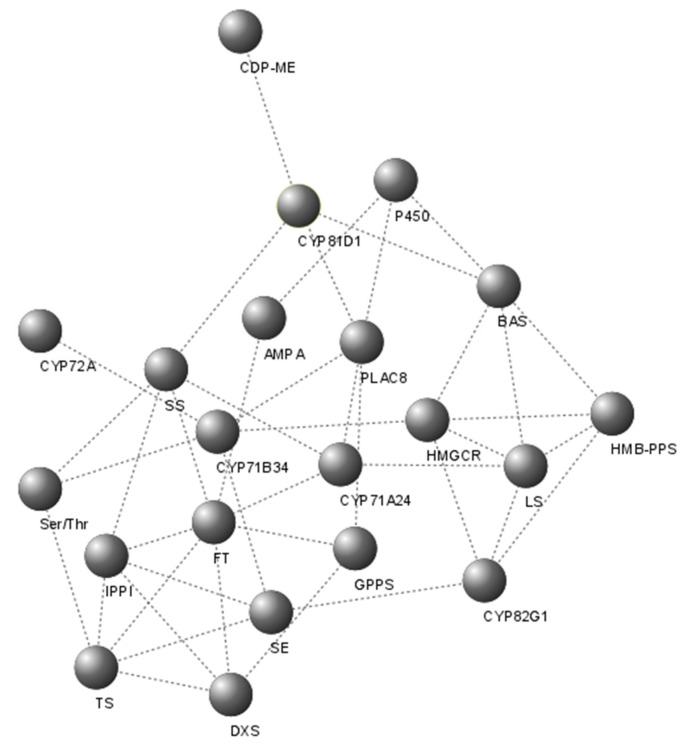
Co-expression network of transcripts related to the synthesis pathway of triterpenes and members of the cytochrome P450 family in leaves of the galphimine producer plants and non-galphimine producer plants. Transcripts are represented by spheres and dotted lines are their interaction. HMGCR (3-hydroxy-3-methylglutaryl-CoA reductase), DXS (1-deoxy-D-xylulose-5-phosphate synthase), CDP-ME (4-diphosphocytidyl-2-C-methyl-D-erythritol kinase), HMB-PPS (4-hydroxy-3-methylbut-2-en-1-yl diphosphate synthase), IPP isomerase (isopentenyl diphosphate isomerase), GPPS (geranyl diphosphate synthase), FT (farnesyl transferase), TS (terpene synthase), SS (squalene synthase), SE (squalene epoxidase), BAS (beta-amyrin synthase), LS (lanosterol synthase).

**Figure 7 plants-11-01879-f007:**
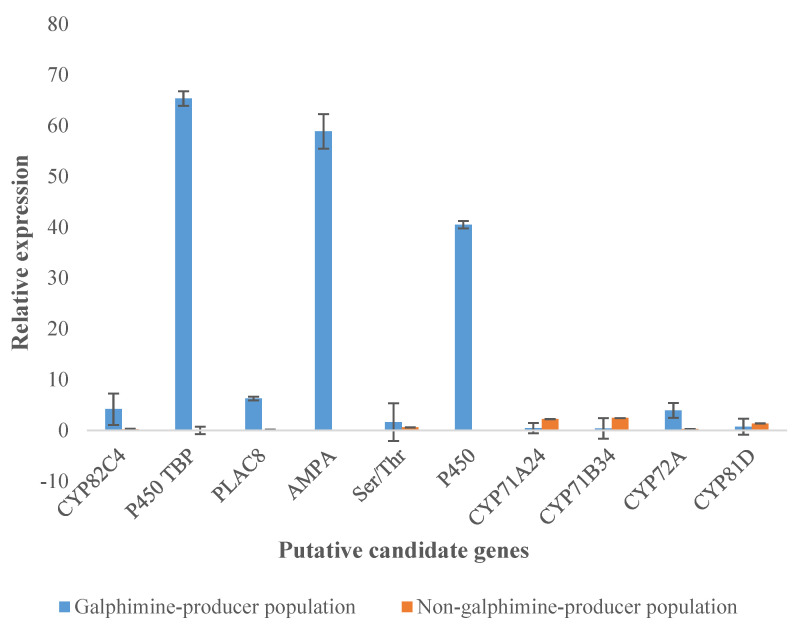
Relative expression of putative candidate genes in leaves of galphimine producer plants (GP) and non-galphimine producer plants (NGP) was calculated by the 2^−ΔΔCt^ method in quantitative PCR analysis. Y-axis represents the fold change value between GP and NGP. AMPA (5’-AMP-activated protein kinase), Ser/Thr (serine/threonine-protein kinase AtPK2/AtPK19).

**Figure 8 plants-11-01879-f008:**
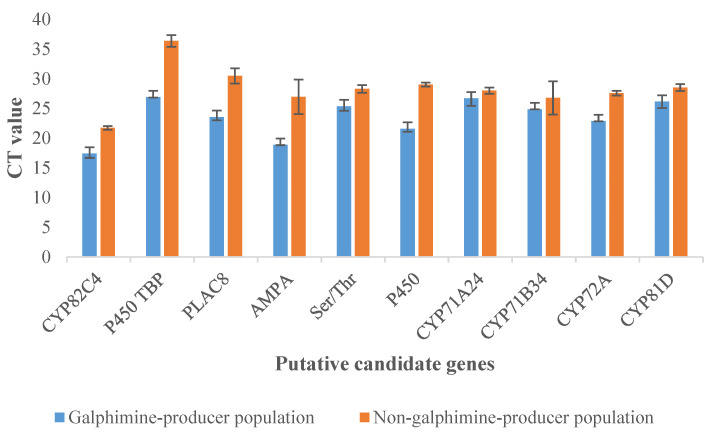
CT values of leaves of galphimine producer plants (GP) and non-galphimine producer plants (NGP) obtained by quantitative PCR analysis. AMPA (5’-AMP-activated protein kinase), Ser/Thr (serine/threonine-protein kinase AtPK2/AtPK19).

**Figure 9 plants-11-01879-f009:**
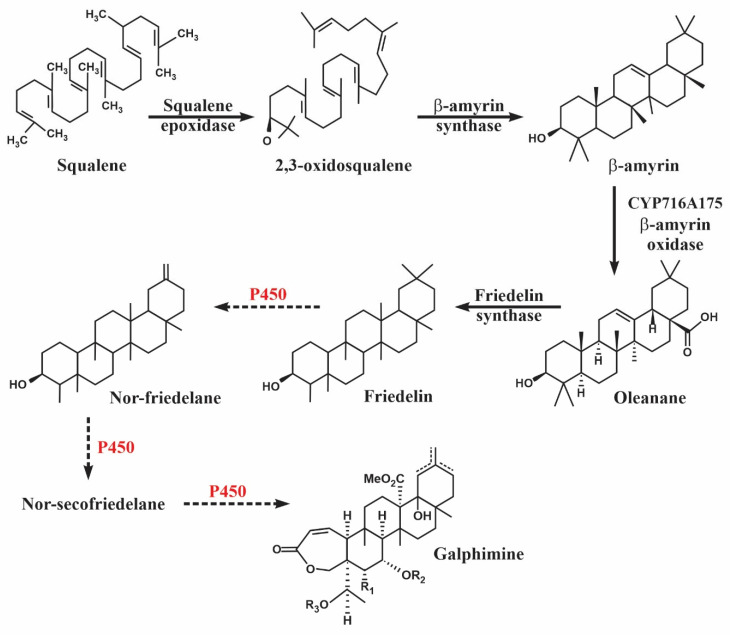
Proposed galphimines synthesis pathway mediated by cytochrome P450 in *Galphimia* spp. plants. Solid arrows indicate pathway reactions identified in previous work [18,28]. Dotted arrows indicate reactions proposed in this study.

**Table 1 plants-11-01879-t001:** Transcripts were obtained after de novo assembly of transcriptomes.

Sample and Replica	Number of Transcripts
Leaves of galphimine producing-population	9023
Leaves of galphimine producing population, replica	8915
Leaves of non-galphimine producing-population	8667
Leaves of non-galphimine producing-population, replica	8896
Roots of galphimine producing-population	5004
Roots of galphimine producing population, replica	4667

**Table 2 plants-11-01879-t002:** Annotation and gene counts of transcripts related to terpene synthesis of two populations of *Galphimia* spp.

	Galphimine-Producer Population		Non-Galphimine-Producer Population		Control	
Annotated Transcripts	Number of Annotations	Gen Counts	Number of Annotations	Gen Counts	Number of Annotations	Gen Counts
3-hydroxy-3-methylglutaryl-CoA reductase	2	13	1	8	1	6
1-deoxy-D-xylulose-5-phosphate synthase	4	15	4	29	-	-
4-diphosphocytidyl-2-C-methyl-D-erythritol kinase	4	18	1	10	-	-
4-hydroxy-3-methylbut-2-en-1-yl diphosphate synthase	5	19	5	16	-	-
isopentenyl diphosphate isomerase	1	52	1	17	-	-
farnesyl transferase	1	6	3	4	-	-
geranyl diphosphate synthase	1	14	1	4	-	-
terpene synthase	1	347	3	253	5	78
squalene synthase	1	7	-	-	-	-
squalene epoxidase	1	16	-	-	-	-
beta-amyrin synthase	2	23	4	14	-	-
lanosterol synthase	7	18	6	23	-	-
P450	2	304.444	-	-	1	4
P450 TBP	19	417.183	-	-	-	-
CYP71A24	1	5	2	3	-	-
CYP71A26	-	-	1	14	-	-
CYP71B10	-	-	1	12	-	-
CYP71B34	1	9	-	-	-	-
CYP71D11	-	-	-	-	1	15
CYP72A	2	10	-	-	-	-
CYP72A15	-	-	1	7	-	-
CYP81D1	1	18	-	-	-	-
CYP82C4	1	231	-	-	-	-
CYP82G1	1	41	1	129	-	-
PLAC8	2	12	-	-	-	-
5’-AMP-activated protein kinase	1	30	-	-	-	-
serine/threonine-protein kinase AtPK2/AtPK19	1	6	-	-	-	-

**Table 3 plants-11-01879-t003:** Sequences of synthesized primers for qPCR.

Gene Description	Gene Symbol	Primer Sequence 5′-3′	PCR Product Length (bp)
CYP82C4	CYP82C4	F 5′TCATTTGGTGGGCTAAAAGC 3′ R 5′ACGAGTCCGTTAATGGTTGC 3′	191
P450 TBP	P450 TBP	F 5′AGCTTCGTCGCAAGTGAAAT 3′ R 5′TAGGACGCCTGCGTTATCTT 3′	189
PLAC8	PLAC8	F 5′ACTGTTGATAACCCCGATGG 3′ R 5′TCTCAGGAAGTCCGAACTGG 3′	170
5′-AMP-activated protein kinase	AMPA	F 5′CCCAAGGAAAAGGTTTCACA 3′ R 5′GGCAAAAGCAATGGCTAAGA 3′	229
serine/threonine-protein kinase AtPK2/AtPK19	Ser/Thr	F 5′TTTCAACTGGACCACAAAGG 3′ R 5′CGTTCGGGAAAGTCTACCAA 3′	164
P450	P450	F 5′CCTTGAGGTCATGGCTGAAT 3′ R 5′GCTTCTCTCCAAAGGCACAC 3′	232
CYP71A24	CYP71A24	F 5′CAAACCGGCCTAAATCAAAG 3′ R 5′GCTCTTGTTCATAACTTTCTCAATC 3′	200
CYP71B34	CYP71B34	F 5′CCGTGAGAGAGGCCATTAAC 3′ R 5′GTGCGAAAGATTTGCGTTCT 3′	160
CYP72A	CYP72A	F 5′TTGTTGGCTTTTCGAGGAAT 3′ R 5′AAGCATCAGGAGTGGCAAAC 3′	167
CYP81D1	CYP81D1	F 5′TCGGAGGATTGGACTACGAC 3′ R 5′TTCCGCCATAACATTTCTCC 3′	219

## Data Availability

Not applicable.

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
