# Peer review of "Identification of Putative Candidate Genes from Galphimia spp. Encoding Enzymes of the Galphimines Triterpenoids Synthesis Pathway with Anxiolytic and Sedative Effects"

_plants, 2022, doi:10.3390/plants11141879_

Round 1
Reviewer 1 Report
You need to correct the form and English of the article: references in paper are written incorrectly, captions to images must to be more precise.
there are many grammatical and spelling errors in the text
Author Response
Reviewer 1: You need to correct the form and English of the article: references in paper are written incorrectly, captions to images must to be more precise. there are many grammatical and spelling errors in the text.
Comments by Authors: We highly appreciate the time and valuable revision provided to our MS by the distinguished Reviewer. The English language throughout the MS has been revised and edited accordingly. Captions of images have been corrected and precised. Grammatical errors and references have been also corrected accordingly.

Reviewer 2 Report
The authors analyzed the transcriptomes of the GP and NPG populations and identified genes that presumably encode enzymes involved in the synthesis of modified triterpenoid galphimines.
The research was carefully planned and implemented. The results' analysis and comparison with those obtained by other authors do not raise any objections.
Author Response
Reviewer 2: The authors analyzed the transcriptomes of the GP and NPG populations and identified genes that presumably encode enzymes involved in the synthesis of modified triterpenoid galphimines. The research was carefully planned and implemented. The results' analysis and comparison with those obtained by other authors do not raise any objections.
Comments by Authors: We highly appreciate the time and valuable revision provided to our MS by the distinguished Reviewer. We will be glad to provide any further detail if required. Thank you.
